# Using geographic information systems to link population estimates to wastewater surveillance data in New York State, USA

**Dustin T. Hill** *, **David A. Larsen**

Department of Public Health, Syracuse University, Syracuse, New York, United States of America

* dthill@syr.edu

## Abstract

Sewer systems provide many services to communities that have access to them beyond removal of waste and wastewater. Understanding of these systems' geographic coverage is essential for wastewater-based epidemiology (WBE), which requires accurate estimates for the population contributing wastewater. Reliable estimates for the boundaries of a sewer service area or sewershed can be used to link upstream populations to wastewater samples taken at treatment plants or other locations within a sewer system. These geographic data are usually managed by public utilities, municipal offices, and some government agencies, however, there are no centralized databases for geographic information on sewer systems in New York State. We created a database for all municipal sewersheds in New York State for the purpose of supporting statewide wastewater surveillance efforts to support public health. We used a combination of public tax records with sewer access information, physical maps, and municipal records to organize and draw digital boundaries compatible with geographic information systems. The methods we employed to create these data will be useful to inform similar efforts in other jurisdictions and the data have many public health applications as well as being informative for water/environmental research and infrastructure projects.

## 1 Introduction

Wastewater from a community provides a wealth of information about that population including types of infectious diseases circulating, types of drugs consumed, and types of food eaten [1, 2]. Wastewater-based epidemiology (WBE) is the use of wastewater to understand public health issues through testing wastewater samples and then estimating community-level trends [1, 2]. WBE became exponentially important during the emergence of COVID-19 with governments using wastewater surveillance to track the SARS-CoV-2 virus to inform disease spread, intensity, and genetic variation [3]. Beyond COVID-19, WBE has a strong record of informing our understanding of antimicrobial resistance [4], expanding the study of endemic diseases, such as influenza and HIV, and guiding public health research and policy into the future [5]. In addition to providing information on infectious disease threats to public health

**Data Availability Statement:** Data used to draw boundaries and described in this manuscript are available upon request to the author. Links to

publicly available databases used in this project are provided in the reference section and supplemental documents. R code for creating U.S. census intersections and an interactive map to view the sewershed data are publicly accessible at https://dthill196.github.io/NY-Sewershed-Populations/. The final GIS shapefile created for this project is publicly available on ArcGIS Online: https://www.arcgis.com/home/item.html?id=e795007660ae4a1fae5f86b40d065b3a.

**Funding:** This project was made possible by the CDC's Environmental Public Health and Emergency Response Program (NYS Unique Federal Award Number NUE1EH001341, NYS Environmental Public Health Tracking Network Maintenance and Enhancement to Accommodate Sub-County Indicators, to D.L.). The funders had no role in study design, data collection and analysis, decision to publish, or preparation of the manuscript.

**Competing interests:** The authors have declared that no competing interests exist.

[6], testing wastewater can inform understanding of the burden of opioid and other drug use in a community [7]. Linking the population to the wastewater sample is a key step in WBE [8, 9], as accurate population estimates are needed to understand what is found in wastewater in terms of population-level burden and create statistical models that use these data [10]. Linking the sample to the population is also necessary for precise public health intervention if an outbreak is detected [11].

The area from which a treatment plant receives wastewater is known as the sewershed, defined as an area of land where all the sewers flow into one point [12]. Sewersheds are usually polygon shapes inclusive of the property parcels that have connections to the sewer system but can also include larger portions of land extent that may contribute runoff rainwater to the system [13]. Sewersheds follow natural hydrology (like rivers and streams), flowing downhill congruent with topography in most cases or with the addition of pump stations. There are, however, sewer systems that "jump" rivers with pipes crossing the banks aboveground or going underneath the waterway. Systems can also include what are known as force mains, which are sewer lines that force water uphill to the final collection system with the assistance of lift stations that feed gravity flow sewers from higher starting elevation [14]. This variation complicates the estimation of boundaries for sewer systems when survey maps do not exist, but there are ways to approximate boundaries such as using municipal borders and tax registries that collect information on whether a parcel of land is connected to public sewer or not [15]. The sewershed boundary can also be linked with other geographically-referenced data, such as population census data, to estimate the population within a sewershed [16], and linkages can be made with environmental data, such as precipitation and temperature, to better understand wastewater test results. Linking wastewater samples to geographically relevant data can enhance research on infectious disease transmission and improve the utility of data related to the social determinants of health.

In New York State, geographic data on sewer systems, including manhole locations, sewer lines, and related infrastructure, are usually maintained by the municipality that owns the system or the engineers that designed it originally (DEC, personal communication). In other words, centralized databases for sewers across different jurisdictions in New York are unavailable (NY GIS, personal communication). As New York State's wastewater surveillance network increased in scale and scope, a centralized database for all of New York State's municipal sewer systems was needed to link testing of municipal wastewater to upstream populations for public health surveillance of SARS-CoV-2, other pathogens, and substances of public health interest. Since no centralized database existed beyond a list of the point locations for each treatment plant per discharge permitting records [17], creation of a database of sewersheds was necessary for the state's wastewater surveillance network. The final product is a spatial database that has every municipal sewer system in New York State mapped and linked to the treatment plant and population served by that sewer system. In what follows, we detail the methods for gathering and drawing the final sewershed polygons and steps for data validation and quality assurance. These methods can be applied in other jurisdictions with similar needs, and we explore some of the uses for these date beyond public health.

## 2 Data and methods

### 2.1 Data sources

**2.1.1 Treatment plant operators.** The NY Department of Environmental Conservation (DEC) provided a database listing 638 municipal wastewater treatment plants [17]. This list included the name of the facility, its location via street address and geocoordinates, as well as other metadata such as permitted discharge capacity. The list of these plants were the targets

for this project with the goal being to find and draw sewershed boundaries for each one. Contact information for 385 of the plants was provided by the DEC, and a survey was distributed to these plants via email and phone in 2021 (see Hill et al., 2022 for the full question list). The purpose of the survey was to determine what data the treatment plant operators had regarding the service area of the facility and what they knew about WBE methods for tracking substances in wastewater for public health. Results on operator knowledge regarding WBE has been published previously [18]. If no data existed, the operator provided a description of the service area to facilitate drawing of the boundary using other data sources. The respondents were followed up with email and phone calls to request they send the data for inclusion in the database. Data were received in different formats including digital shapefiles of the boundary, sewer manholes, and/or sewer mains; physical maps of the boundary, sewer manholes, and/or sewer mains; and address lists of the properties connected to the sewer system.

**2.1.2 New York state parcel database.** NY maintains a geodatabase of property tax data for all properties in the state. Data for many counties is publicly available with others accessible upon request. We received permission from the state Geographic Information Systems (GIS) office for use of the data to aid in construction of the sewershed boundaries. The tax parcel data consisted of spatial polygons and included a field indicating if the parcel was connect to apublic sewer or a private system. This allowed classification of parcels based on sewer type. In addition, special district tables were provided by the state GIS office as companion material to the tax parcel data and they included information on water and sewer districts that could be joined to the tax parcels by parcel ID number. The sewer district data was not complete for all counties, but, for counties where it was available, it was used to classify separate sewer systems that were adjacent. The version of the data used was from 2020 [19].

**2.1.3 Municipal websites and other data sources.** In addition to calling municipalities, we researched public sewer systems online through town, village, and city websites searching for any published records, reports, or information on the location of sewer service areas. From public websites, we were able to obtain copies of physical maps and descriptions of some sewer systems. A list of websites for source material is provided in the supplementary material in S1 Appendix.

For the three counties in northern New York, additional data were provided by the Development Authority of the North Country (DANC), which is a public benefit corporation set up to manage the infrastructure needs for those counties [20]. DANC provided estimates for 55 sewershed boundaries based on work they completed using village boundaries, sewer billing data, and sewer district data. Further description is available in S1 Appendix.

## 2.2 Combination of data

Data were collected and compiled for each county in NY using eight methods listed in Table 1. Wastewater treatment plants (WWTPs) were mapped in the county to provide the initial locations for communities that would have access to public sewer. The first sewersheds that were added were those that were provided by the treatment plant or municipal/county officials, and these were not modified. Following these, if we had an address list for properties that were billed for public sewer by the municipality, we matched parcels from the NY tax parcel data for the county. The boundaries for the parcels were then dissolved to form the final boundary.

For municipalities that we received digital manhole or sewer main data, we mapped those against the tax parcel data for the county and selected the parcels that were adjacent to sewer mains and proximate to manhole locations (Fig 1A and 1B). These parcels were then dissolved to form the sewershed boundary and linked to the municipality's treatment plant. The other type of data provided were physical maps showing the boundaries of the service area. For these

**Table 1. Methods for digitizing sewersheds.**

| Method | Method description | Count | Percent | Median flow (mgd) | Mean flow (mgd) |
|---|---|---|---|---|---|
| **Digitized from address list** | Digital shapefile boundary for the sewershed was provided by a third party (i.e., the treatment plant operator or owning municipality). | 1 | 0.2 | 0.09 | 0.09 |
| **Digitized from manhole/ sewer main shapefile** | NY State Tax Parcel database was used to estimate parcels served by this sewer system. Parcels that are cross listed on the address list provided were assigned to the sewershed to draw the boundary. | 35 | 5.8 | 0.40 | 1.93 |
| **Digitized from physical map (JPEG/PDF)** | Manhole and or sewer main data were sent by the treatment plant operator and the boundary for the service area was created using these data and NY State Parcel data that intersected with the manhole/sewer main data. | 126 | 21.0 | 0.52 | 3.77 |
| **Parcel digitized from DANC records** | A document containing a map of the region served by the treatment plant was used to draw the boundary using ArcGIS software. | 55 | 9.2 | 0.10 | 1.26 |
| **Parcel digitized from description** | Development Authority of the North County (DANC) provided a shapefile of sewersheds that were created starting in 2010 and updated annually. The version this project used was obtained in 2021. Boundary matches current estimates and therefore DANC boundary was used for this treatment plant's sewershed. | 19 | 3.2 | 0.99 | 3.30 |
| **Parcel digitized no description** | Treatment plant operator provided a text description of the service area including roads, natural boundaries, number of influent points, and towns/villages/cities served. The boundary was drawn using ArcGIS software and Census data for major roads, municipal boundaries, and NY State Tax Parcel database. Tax parcels that are recorded as paying a sewer tax and fall within the described service area were assigned to this sewershed. | 304 | 50.8 | 0.35 | 1.34 |
| **Provided by treatment plant/municipality/county** | New York State Tax Parcel database was used to estimate the parcels served by this sewer system. Parcels that are recorded as paying a sewer tax and proximate to the wastewater treatment plant's geolocation were assigned to the sewershed. Adjacent sewer systems were subdivided using sewer district data when available. | 58 | 9.7 | 11.00 | 47.14 |
| **Village boundary** | No adjacent parcels were indicated to be on sewer. Municipal village boundary was used as a placeholder. | 1 | 0.2 | 0.05 | 0.05 |
| **Total number of sewersehds** | | 599[1] (596 WWTPs) | | | |

**Note:** Of the original 638 treatment plants identified in DEC's database, sewersheds were drawn for 592 of them with 4 additional identified that were not in the database. There were 46 that we did not draw sewersheds for because they either do not exist anymore or are included in another system.

maps, we aligned the borders of the map with the tax parcel data for that county and municipality and then selected the parcels within the boundary (Fig 1C). Listed roads and other landmarks that formed the boundary were used to ensure the correct parcels were selected. The parcel boundaries were then dissolved to form the final sewershed boundary.

Boundaries provided by DANC were compared to the reported parcels on public sewer. For many municipalities, the boundaries corresponded well and the DANC boundary was used, however, in some cases adjacent parcels were reported to be on public sewer but not in the DANC sewershed. Therefore, we added these additional parcels to form the final sewershed combining the DANC sewershed with adjacent parcels on public sewer for the same municipality (Fig 1D).

For the remaining municipalities that we did not have data provided or could not find public records for sewer boundaries or districts, we drew the boundaries based on the parcels on public sewer near the geolocation of the municipal WWTP (Fig 1E). For many locations, the clusters of sewered parcels corresponded well to the locations of treatment plants and there were clear boundaries between sewer systems. For some counties with several WWTPs serving adjacent communities, it was harder to distinguish between the sewer parcels and link to the correct WWTP.For those locations, we examined the sewer district data from the tax parcel special district tables. Where available, sewer districts were identified and grouped together to

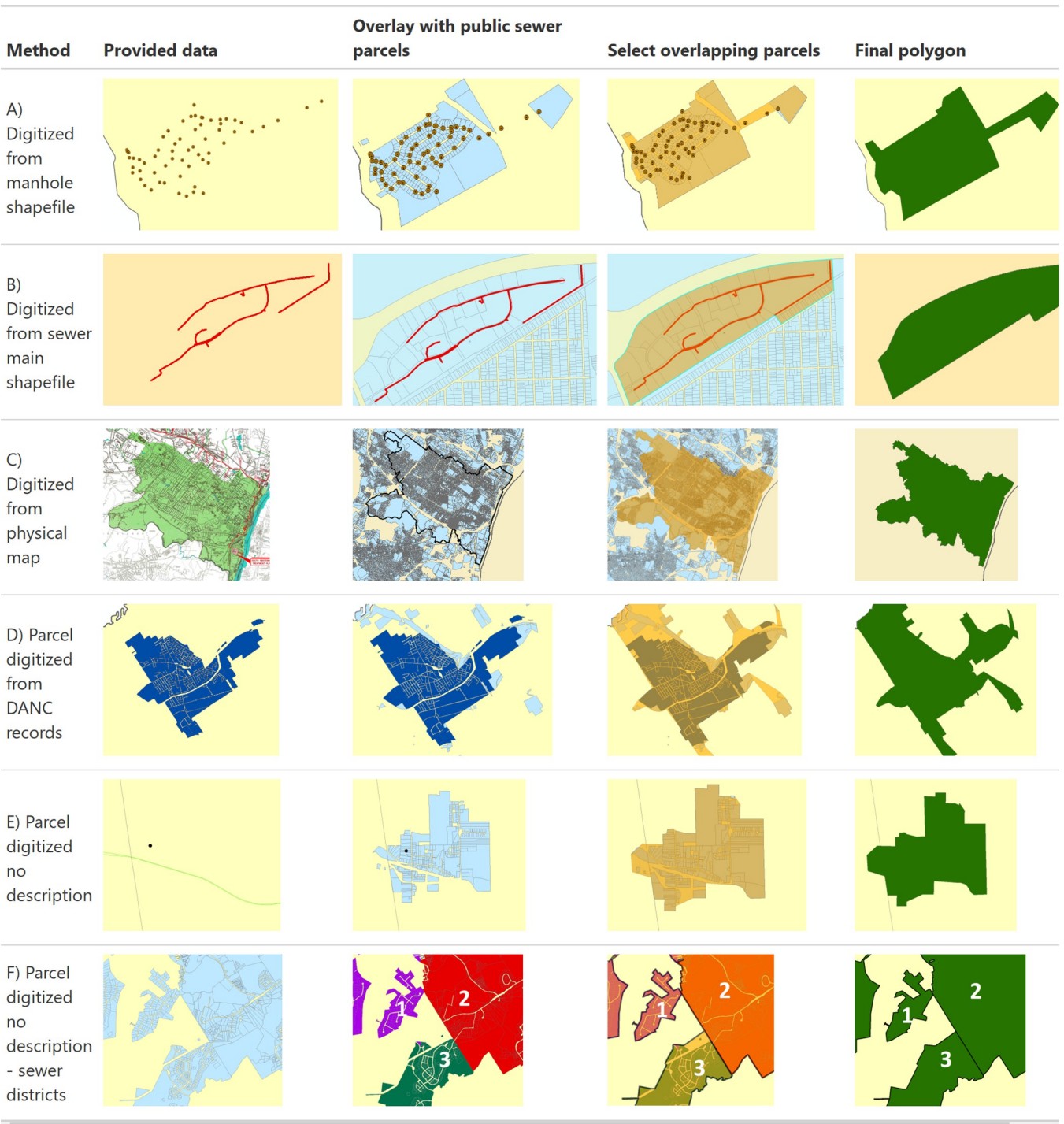

**Fig 1. Methods used to create sewersheds.** Base geography layers retrieved from U.S. Census TIGER/Line shapefiles https://www.census.gov/geographies/mapping-files/time-series/geo/tiger-line-file.html and NY State Tax Parcel data http://gis.ny.gov/parcels/.

form distinct sewer systems that were then linked to the correct WWTP (Fig 1F). Some counties did not have good sewer district data and the treatment plant or municipal official was contacted to request information on the boundaries for adjacent sewer systems.

## 2.3 Adding population data

Estimates for 2020 population by sewershed were calculated using R statistical software version 4.1.1 [21]. We first estimated the 2010 population for each sewershed using an overlay of 2010 United States (U.S.) Census blocks on top of the sewershed boundaries. Census blocks were selected because they represent the smallest geometry for public census data and increased the accuracy for the intersection with sewershed boundaries. We calculated the proportion of the area for partial block overlap and then assigned the proportional 2010 decennial population of the block to the sewershed assuming equal distribution of the population in the blocks. The apportioned values were then aggregated to obtain a total population estimate for the sewershed.

We then repeated this procedure and the apportionment method using 2010 decennial population data for the block group and 2018 American Community Survey (ACS) data for the block group to get 2010 and 2018 population by sewershed based on block groups. We used these values to estimate the annual rate of population change per sewershed using the following formula:

$$Annual\ growth\ rate = \left(\frac{(2018\ pop - 2010\ pop)}{2010\ pop}\right)/8\ \text{years} \tag{1}$$

We then applied this average annual change to the sewershed population based on the block data from 2010 and estimated the population after ten years of growth using the following formula to calculate 2020 population estimates:

$$2020\ pop\ est = 2010\ pop * (1 + annual\ growth\ rate)^{10} \tag{2}$$

We obtained all population estimates using the "tidycensus" R package [22] and we use the "tigris" package [23] for U.S. Census geometry data. R code for U.S. census intersections is publicly available at https://dthill196.github.io/NY-Sewershed-Populations/

## 2.4 Quality assurance

We evaluated sewershed boundaries by cross referencing data types when multiple forms were available such as having physical maps and shapefiles. For sewersheds where the boundaries were in question or more difficult to estimate, direct contact and consultation with treatment plant operators was used to check boundaries against the knowledge of the treatment plant staff. Short of formal land surveys of the municipality, the data we created are not easily validated. To evaluate sewershed boundaries, we ran correlations of the population estimates, population density, and sewershed area of the final polygons with the permitted discharge capacity. Discharge capacity increases with population served and was one way to check that methods for creating sewersheds produced good estimates for areas served, and the method has been used by other researchers working on population and treatment plant linkages [24]. Discharge capacity was used instead of average daily flow because the former was available for every treatment plant while the latter was only available for 210 locations. Discharge capacity and the population variables were right skewed, and therefore, we used the base ten log transformation of each variable to compare their linear correlation.

## 3 Results

Of the original 638 municipal treatment plants identified using the DEC database, we were able to create boundaries for 592 of them. There were 46 WWTPs in the DEC list that we did not create sewershed boundaries for because they either did not exist anymore or were

additional permits for coincident treatment plants representing combined sewer overflow facilities or large pump stations (see S2 Appendix for a list of permits and reasons they were excluded). In addition, we created sewershed boundaries for 4 additional WWTPs not in the DEC database. Two of these facilities are owned by Native American communities and are on land owned by those communities with their permits supervised by Federal agencies. We included them because their boundaries were evident from tax parcel data for public sewer that were not linked to any treatment plant in the DEC list. Two additional facilities were reported by county governments with permit information and physical maps provided showing their locations. The final count for our database is 596 municipal WWTP sewersheds. Three WWTPs have two sewersheds each representing separate influent streams from different parts of their service area bringing the total number of sewersheds to 599 (Table 1).

We received 116 completed surveys with 19 responses that had a description of the service area. These were then combined with other data sources, such as the NY Tax Parcel database, to draw the sewershed boundary. The remaining 97 responses indicated that the plant had data on the service area of the plant. The survey respondent was then contacted requesting they share the data with our research team. We received digital data for the boundaries of 58 treatment plants and digital data for manholes and sewer mains for 35 treatment plants, and an address list for one treatment plant. Data were retrieved in a combination of survey callbacks and contact with treatment plants after the survey ended (Table 1). In total, 294 sewersheds were digitized using data provided or descriptions of the system representing 49.1 percent of the final database for NY sewersheds and these facilities were among the largest in the state with a median flow rate of 11 million gallons per day (mgd) and mean of 47 mgd (Table 1).The remaining sewersheds (304) were digitized from NY tax parcel data and sewer district data with one sewershed digitized using the village municipal boundary (Table 1). The treatment plants are distributed around the state with some regions and counties having many separate sewer systems with others having fewer (Fig 2). The total estimated state population on public sewer is over 16 million, which is approximately 82.6 percent of the state population (using 19.5 million as the estimated NY population in 2020). Population coverage per sewershed ranged between less than 100 people to as high as 1.2 million. Population density ranged between 1.6 people per km$^2$ and 28,000 people per km$^2$. Sewershed area ranged between 58,000 m$^2$ and 990 km$^2$. There was high correlation between the log permitted discharge capacity of each facility and estimated log population served (Pearson correlation = 0.897, Fig 3), moderately high correlation between log discharge capacity and estimated log population density (Pearson correlation = 0.569, Fig 3), and high correlation between log discharge capacity and log sewershed area (Pearson correlation = 0.852, Fig 3) (note that untransformed correlations were statistically significant with Pearson correlations of 0.952 for population, 0.7624 for population density, and 0.437 for sewershed area). We examined correlations across the different methods for creating sewersheds (Table 2) finding most Pearson correlations were high with a few exceptions. The use of manhole and sewer main data was correlated significantly with population and sewershed area, but for population density, we found lower correlation with a Pearson estimate of 0.195 and p-value = 0.27 (Table 2). When comparing number of sewers between urban and rural counties, urban counties have slightly more sewers with a median of 9 versus a median of 7 in rural counties (Table 3). A greater difference is evident when comparing total population on sewer with rural counties having a median of over 17,000 people on sewer while urban counties have over 160,000 people on sewer (Table 3). Further, the WWTPs in urban counties tend to be larger with a median discharge capacity of 500,000 gallons per day with rural counties median capacity being 350,000 gallons per day. The proportion of population served by sewer ranged from 5.3% in Hamilton County (a rural county in the Adirondack Park) to 100% in the New York City counties.

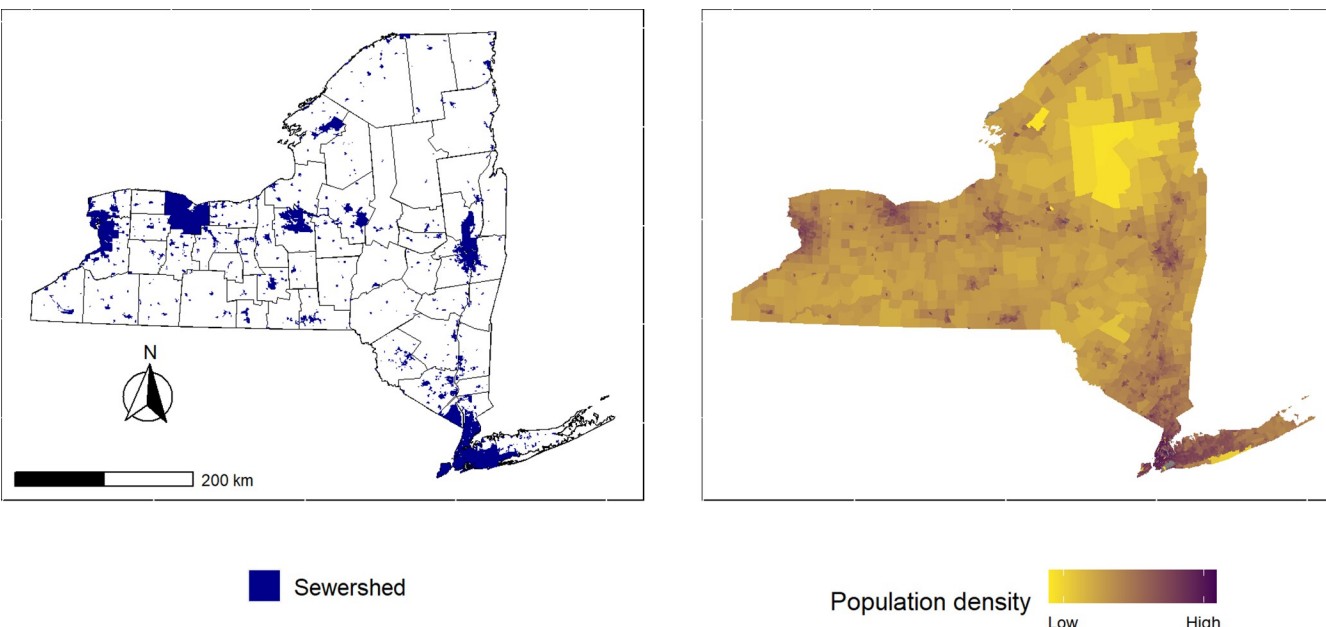

**Fig 2. Spatial distribution of sewersheds in New York.** Highly sewered counties and regions correspond with population centers across the state. Base layer for New York retrieved from U.S. Census TIGER/Line shapefiles https://www.census.gov/geographies/mapping-files/time-series/geo/tiger-line-file.html.

## 4 Data availability

Data used to draw boundaries and described in this manuscript are available upon request to the author. Links to publicly available databases used in this project are provided in the reference section and supplemental documents. R code for creating U.S. census intersections and an interactive map to view the sewershed data are publicly accessible at https://dthill196. github.io/NY-Sewershed-Populations/. The final GIS shapefile created for this project is publicly available on ArcGIS Online: https://www.arcgis.com/home/item.html?id= e795007660ae4a1fae5f86b40d065b3a.

## 5 Discussion and conclusions

### 5.1 Evaluation of sewershed boundaries

Correlation results between the final sewershed polygons, the log of discharge capacity of the WWTPs, and the three metrics of log population, log population density, and log sewershed area were high for all methods used (Fig 3) suggesting that each method contributed acceptable results. Further, there were differences correlation between data sources based on treatment plant size with those drawn from physical maps coming from the largest plants (median capacity 10 mgd) and the smallest plants requiring greater reliance on tax parcel data (Table 2) One exception to the high correlations were sewersheds drawn using manhole or sewer main shapefile data (Table 2). This method resulted in the lowest correlation between log discharge capacity and log population density, however, there was high correlation for the log of estimated population and log sewershed area for this method. Therefore, despite the one low value for this method, we feel that the resulting sewershed boundaries are acceptable estimates for the service areas of the treatment plants. There were 35 sewersheds drawn using this method with the majority (n = 21) from one county. Removing the county in question from the correlation test yielded a new correlation coefficient of 0.671 suggesting there might be greater uncertainty with the sewershed boundaries from that county.

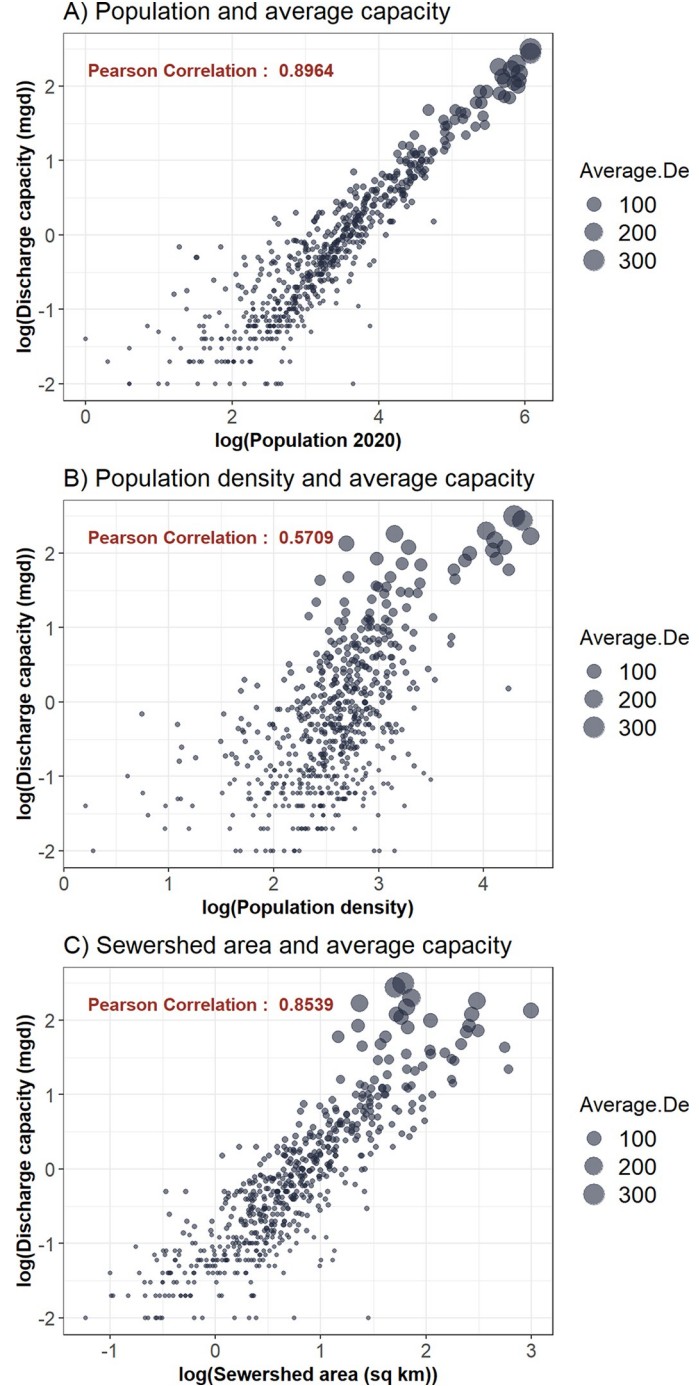

**Fig 3.** Scatterplots between log permitted discharge capacity of WWTPs and A) log estimated population, B) log population density, and C) log sewershed area.

The county in question is unique in being highly populated but most of the county population not living on public sewer and, instead of the more typical private septic system for residents, most of the public wastewater goes into what are known as cesspools [25]. The county in question has a population over 1 million people but our estimates report only 397,000 of the residents are on public sewer meaning it has high population density but low public sewer

**Table 2. Correlation results for each method.**

| Method | Median discharge capacity (mgd) | Discharge capacity and estimated population | | Discharge capacity and population density | | Discharge capacity and sewershed area | |
|---|---|---|---|---|---|---|---|
| | | Pearson correlation | P value | Pearson correlation | P value | Pearson correlation | P value |
| Provided by treatment plant/municipality/county | 0.36 | 0.83 | 0 | 0.439 | 0 | 0.826 | 0 |
| Parcel digitized no description | 0.52 | 0.876 | 0 | 0.514 | 0 | 0.853 | 0 |
| Parcel digitized from description | 0.99 | 0.974 | 0 | 0.615 | 0.005 | 0.853 | 0 |
| Digitized from physical map (JPEG/PDF) | 10 | 0.972 | 0 | 0.673 | 0 | 0.698 | 0 |
| Digitized from manhole/sewer main shapefile | 0.4 | 0.707 | 0 | 0.195 | 0.27 | 0.668 | 0 |
| Parcel digitized from DANC records | 0.1 | 0.897 | 0 | 0.544 | 0 | 0.822 | 0 |

population, which might lead to the lower correlation between log population density and log discharge capacity of the sewer systems. In addition, the county has 30 WWTPs with 22 plants with discharge capacities below 1 million gallons per day. Therefore, the county might be an anomaly among those in NY with regards to public sewer systems. Despite the low correlation for population density, the county has high correlation for population estimates and sewershed area with discharge capacity suggesting the resulting boundaries are comparable in their estimates to the rest of the state.

While there are many differences between urban and rural counties in terms of the size of treatment systems and the number of people on sewer (Table 3), there are not many more plants in urban counties than there are in rural counties. Urban counties had a total of 331 distinct systems and rural counties had 264 (Table 3). This could be due to the old infrastructure in New York where many communities have separate systems that have not been combined as communities expanded [26]. Some of the most urban counties had the highest number of sewer systems such as Suffolk County with over 1.4 million people and 30 distinct sewer systems (see S1 Table for more county specific details).

The resulting database of sewersheds includes 4 treatment plants not in the DEC database. These additional plants were drawn based on information from the municipalities and because there were public sewer parcels that matched the description by the municipal officials. DEC's public database could be behind in its updates between discharge permit documents and adding locations with the most recent evaluation of sewer infrastructure from 2008 [26]. We also did not create boundaries for 46 treatment plants listed in DEC's database for several reasons including that they do not exist anymore, or they are included in the sewershed of another facility because they are a smaller pump station or overflow facility. A complete list of the facilities not included, and explanations why are provided in the supplemental material.

## 5.2 Applications for public health, research, and policy

The NY Sewershed database was created to support public health efforts related to WBE including the important role of providing population estimates for wastewater test results [10,

**Table 3. Summary of sewershed populations for rural v. urban counties.**

| | Median number of sewers | Median sewered population | Median proportion of county population on sewer | Number of total sewers | Median sewershed population density | Median sewershed area (sq km) | Median WWTP discharge capacity |
|---|---|---|---|---|---|---|---|
| Rural | 7 | 17,803 | 0.351 | 264 | 289.391 | 3.923 | 0.350 |
| Urban | 9 | 167,687 | 0.700 | 331 | 494.689 | 4.797 | 0.500 |

16]. Linking wastewater results to the upstream population is essential for interpretation of findings [6] particularly for case and incidence data for health outcomes of concern such as COVID-19 [27], Polio [28] and other enteroviruses, and opioid use [29]. Community-based research that uses wastewater data needs accurate estimates for population served by the facility and large-scale investigations that need to scale up quickly may not have the time to create boundaries or provide estimates in a timely manner particularly if the data are to be used for public health policy. Linking wastewater surveillance data to communities is necessary for policy applications of WBE with increasing application in the U.S. since the start of the COVID-19 pandemic [3, 30]. Beyond WBE, this database could be used to support water quality research, hydrology research, and policy related to infrastructure improvements.

Water quality researchers might find utility in these data examining freshwater systems and discharge locations. With the sewershed boundaries, researchers now have estimates for where sewer systems border or even cross waterways enabling inclusion of these data in research plans as has been done before [9]. In addition, hydrology research can use these data to understand the flow of wastewater to treatment plants from communities and potential relationships with topography and hydrography [31]. Lastly, the infrastructure needed to build and maintain sewer systems is aging in NY according the DEC [26] with 36.2 billion dollars estimated to repair, replace, or update sewer systems. Comprehensive mapping of the served communities through the NY sewershed database might be helpful in state-wide and county-wide planning for consolidating sewer systems. While not an official survey of sewered communities or systems, the database we created provides an important step in identifying current sewer systems in NY.

## 5.3 Advice for adapting these methods for other jurisdictions

Accuracy of the boundaries was not easy to determine because there was no pre-existing data for many locations to compare. However, the boundaries and physical maps showing the borders of sewer systems that did exist and were provided to our research team gave us a set of known examples that showed what sewer boundaries looked like and how they followed some features in the built environment like roads. Further, working directly with WWTP operators was essential for understanding many sewer systems and the areas they served. Without the cooperation of the operators, many smaller sewersheds might not have been drawn. We encourage similar projects in other jurisdictions to build relationships directly with WWTP engineers because their knowledge will be essential in drawing the boundaries for their systems. The other data source that assisted with determining if a boundary was accurate or not was if all public sewer parcels from the tax parcel database were accounted for in a county once boundaries were drawn. There were instances where a county had sewer parcels in small clusters that did not adjoin others making it difficult to match them. These required direct consultation with local officials, but most counties did not have any of these "wayward" parcels meaning that, once boundaries were drawn for the county's sewer systems, all public sewer parcels were accounted for.

In addition, to adapt our methods in other jurisdictions, we recommend that researchers start by identifying the location of WWTPs. In the U.S., state environmental agencies will have record of permitted plants and their location that can be publicly accessed. In other countries, this information might be kept in city or provincial offices and may not be as easy to obtain. Still, a global mapping effort did find sewer system data for many countries to be publicly available, which can provide a starting point [24]. Once a list of WWTPs is obtained and the researchers identify the ones that they would like to map (if not mapping all of them), the next step is to prepare a protocol for contacting each plant's management or operations team. For

our project, a contact list was provided that contained information for most but not all operators. If contact information was not available, our team took the next step of calling village and town clerks to obtain contacts for the local treatment plant. This approach may not work in all countries and therefore it might be necessary to directly partner with WWTP permitting agencies or ministries in the study country. Such agencies should know who and how to contact WWTP managers. Once these partnerships and contacts are obtained, creating the data becomes easier. Urban locations tend to have more data than rural locations making mapping rural sewer systems more difficult. These systems benefitted substantially from direct knowledge from treatment plant operators but lacking their input, researchers might consider satellite imagery showing locations of sewer infrastructure in a community such as manholes and use these to estimate sewer service areas.

## 5.4 Limitations and future directions

The total effort to complete this project took 6 months. The survey design, distribution, and collection of responses took about three months with the remaining three months devoted to contacting treatment plants and drawing sewershed boundaries. Completion of ten sewer boundaries in ArcGIS drawn from parcel data took approximately three to six hours depending on the size of the system. Larger systems (larger area) took longer to draw with smaller systems taking less time. Validation and comparison to physical maps also adds time to the process. There is no subset of boundaries that we feel is inaccurate, though the sewershed that was drawn from the village boundary is likely the least accurate because it had no corroborating data. We feel that the most accurate (excluding boundaries provided to us) were systems that had physical maps to reference. The process we took could be improved with a longer amount of time spent reaching out to treatment plant operators since not all contact methods were exhausted. In addition, we did not use satellite data in this project but that might have been useful in validating boundaries by identifying manhole locations.

The sewersheds that we created come with some additional limitations. For all data created, there is uncertainty around the estimated boundaries because they were not produced from formal surveys, but instead methods that estimate the boundary based on secondary data sources. Physical maps that were used might be the most accurate followed by digital data provided by treatment plants, however, the date of some of the maps' original drawing go back several years, sometimes to the original building of the treatment plant. In addition, the tax parcel data are up to date but come with limitations of their own including disclaimers regarding their accuracy. Therefore, these data should not be used in place of official surveys or assessments of sewer systems for the purpose of projects related to infrastructure updates. Instead, these data can support where surveys could be done and help locate sewer systems if previously unmapped. In addition, our evaluation relied heavily on the log transformation of permitted discharge capacity, which is larger than actual average daily flow. This could vary with the age of the system, design of the plant, and whether it is a combined or sanitary system. Despite this, the capacity and average flow were linearly correlated with each other and discharge capacity was a suitable proxy for flow and was readily available. Another limitation is that our sewer boundaries may include properties with private systems or what are known as straight pipes that discharge directly into waterways. We had no way of excluding these in our approach since there was no way to identify all of these "inter-sewer" systems with the resources we had. Future work may require additional time and engagement with local communities to identify these locations.

Further, U.S. Census data estimates represent the population at one time point and data may not be the best match for some locations that are tourist destinations or have frequent

population changes due to seasonal residents. Locations of this kind should use the provided population estimates adjusted for their research endeavor and where necessary seek better estimates from other sources. In addition, low population estimates were obtained for many locations (56 sewersheds had population estimates below 100). These low population levels could be due to the limits built into the U.S. Census data where population numbers may be suppressed due to low number of residents in rural communities. Alternatively, these estimates might be accurate for a few sewer systems that were very small with some serving a single street with a dozen homes. Similar to sewersheds with population fluctuation, alternate sources may be needed to improve estimation for small sewersheds such as seeking information directly from municipalities on number of sewer hookups.

The future of these data will require updates as time progresses with the potential for sewer systems to be consolidated and new systems to be built making current estimates less reliable. Updates can be done using revised tax parcel data showing additional parcels on sewer as well as surveys of treatment plant operators and municipalities to identify any changes. Updates for the New York data will not have to start from scratch and can use our data as a starting point. Methods described in this paper can be applied in other jurisdictions including surveying treatment plants for data availability and working with public and private repositories of land use and property data to estimate sewershed boundaries. In addition, these methods can be used to aid in formulation of upstream and community-level sampling approaches for public health where sewersheds are sub-divided into smaller units to better identify outbreaks. These smaller geographic units might also incorporate additional data like sewer travel time because that can have an impact on the substances tested in wastewater. Also, future projects might seek to include industrial discharges or systems in their mapping to disentangle the contribution of industrial wastewater from the community contribution. This might require geolocating industrial facilities or working with permitted, private facility data that state environmental agencies should have on record. The potential for changes in sewer infrastructure is a challenge, but the current dataset and methods are valuable in their comprehensiveness, novelty, and utility to support public health, water research, and policy.

## Supporting information

**S1 Appendix. Details on boundary creation methods.**
(DOCX)

**S2 Appendix. Excluded permits from the final database with explanations.**
(XLSX)

**S1 Table. Table summarizing sewershed results for each county in New York.**
(DOCX)

## Acknowledgments

We would also like to thank Ed Hampston, AJ Smith, and Eric Weigert for their contributions regarding WWTP locations and contact information. Also, a thank-you to Hannah Cousins, Bryan Dandaraw, Catherine Faruolo, Alex Godinez, Sythong Run, Simon Smith, Megan Willkens, and Shruti Zirath for their help calling treatment plants. Additional thanks to Kate Kiyanitsu from the NY GIS office for providing access and support for NY tax parcel data. Thank-you to Tabassum Insaf, Abigail Stamm, and Jeff Bryant from NY DOH for helping refine some Sewershed boundaries. Thank-you to Mary Collins for connecting us with the NY GIS office. Finally, thank-you to Star Carter at the Development Authority of the North Country for providing estimates for municipal sewer districts for several treatment plants.

## Author Contributions

**Conceptualization:** Dustin T. Hill, David A. Larsen.

**Data curation:** Dustin T. Hill.

**Formal analysis:** Dustin T. Hill.

**Funding acquisition:** David A. Larsen.

**Investigation:** Dustin T. Hill.

**Methodology:** Dustin T. Hill.

**Project administration:** David A. Larsen.

**Resources:** David A. Larsen.

**Software:** Dustin T. Hill.

**Supervision:** David A. Larsen.

**Validation:** Dustin T. Hill.

**Visualization:** Dustin T. Hill.

**Writing – original draft:** Dustin T. Hill.

**Writing – review & editing:** Dustin T. Hill, David A. Larsen.

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
