## [Decision Letter · Decision Letter 0]

31 Oct 2022

PGPH-D-22-01363

Using geographic information systems to link population estimates to wastewater surveillance data in New York State, USA

Dear Dr. Hill,

Thank you for submitting your manuscript to PLOS Global Public Health. After careful consideration, we feel that it has merit but does not fully meet PLOS Global Public Health’s publication criteria as it currently stands. Therefore, we invite you to submit a revised version of the manuscript that addresses the points raised during the review process.

We look forward to receiving your revised manuscript.

Kind regards,

Thomas P. Van Boeckel

Academic Editor

Journal Requirements:

1. Please provide separate figure files in .tif or .eps format only and remove any figures embedded in your manuscript file. Please also ensure that all files are under our size limit of 10MB.

2. We have noticed that you have uploaded Supporting Information files, but you have not included a list of legends. Please add a full list of legends for your Supporting Information files after the references list. 

Additional Editor Comments (if provided):

Reviewer #1

This is a sampling protocol paper on how to use GIS files of sewer systems across jurisdictions for use in defining sewersheds for public health sampling, with a case study of New York presented. The paper is well written, practical, and innovative in documenting this approach.

1.) Abstract. Some areas of the United States do have centralized databases for geographic information on sewer systems. Do any other states have a fully centralized database? Add a qualifier to the statement “there are no centralized databases for geographic information on sewer systems”

2.) Line 40, add at the end of this line “or with the addition of pump stations.”

3.) Line 51, again need to qualify that you are talking just about NY here.

4.) Was there any overlap in service areas identified? This could be in high precipitation events for example where one sewershed diverts to another one.

5.) Were any of your sewersheds CSO/SSO?

6.) Authors should calculate high and low ranges for the counties studied of how many residents are on sewer, which indicates the total portion of the community captured by WBE samples.

7.) I think one of the most useful parts of this work is not the population covered, but knowing the spatial “gap” in households without services in areas where it “should” be available. For example, this might be an old house with a failing septic tank or straight pipe, which in addition to excluding this house from WBE samples also poses a risk to human health and the environment of surface water and your system could highlight these houses for public health attention. Can you choose one or two areas and conduct this additional analysis for comparison?

8.) In many older mid-western cities, straight pipes remain in pockets of urban areas. How do you account for this in NY, if present.

9.) A table in the supplement giving an overview of population size and a list of the counties you were/were not able to include would be helpful.

10.) Conclusion. How could the findings of this work be implemented across 2 or 3 jurisdictions? Is there a list of minimum information needed for this merge (Add a table)? What can be done within rural environments with limited digitization of sewer pipes (physical maps are only covered in the limitations)?

11.) Finally, when you submit the corrected version, please do check thoroughly, in order to avoid grammar, syntax or structure/presentation flaws.

Reviewer #2

This is a very useful paper detailing a project in NY state delineating the boundaries of sewersheds using a variety of extant data. This paper would be very helpful for practitioners and researchers and the resulting sewershed data has been valuable for public health actions and research. The manuscript is clear and the methods area well described; Fig 1 is very nice.

I have two primary comments. First, I do not agree that the correlations between log(capacity) and population metrics provide a high degree of evaluating accuracy. I have provided more detailed comments below. If the correlations are with capacity and not log(capacity), I apologize for my misunderstanding.

I imagine that the authors have a great deal of knowledge about how accurate the resulting sewersheds are and the factors that result in having accurate or inaccurate boundaries. This leads to my second comment: this experience and a few other topics would be of great use to readers and should be covered in he discussion.

I have provided more detailed comments addressing these two points below.

Specific comments.

Background

Line 37 – Consider replacing ‘area’ to ‘extent’

Line 42 – Note that lift stations are also used to not only feed force mains, but to feed gravity flow sewers from a higher starting elevation.

Methods

Line 84 – where did the special district tables come from?

Line 143: Did you use the same dasymetric apportionment for he block groups as you used for the 2010 block populations? Please clarify.

Line 154: It doesn’t seem that this process was validation as you didn’t appear to have a ‘true’ or gold standard boundary to compare to other derived boundaries, and when you did have a ‘good’ boundary, you used that. In some cases it sounds like you used a combination of methods. In addition it appears that you compared methods to get a sense of the reliability of your estimates. Comparing population estimates, which have inherent and non-random biases due to the apportionment methods, to the permitted flow may be a form of validation. However permitted flow is not actual average flow (which we have been able to get from most WWTPs). Were there any criteria used to determine if a sewershed appeared to be accurate based on a relationship between population and permitted flow?

This is a relatively small point. I recommend that you re-word the section and characterize it as just QA. Reliability is a concept you seem to be describing in this section. Quality Control is more accurate as you are assessing the resulting product (the sewershed boundaries) rather than the efforts and protocols set up in the process of creating the sewersheds to assure high quality boundaries.

Results

Line 170. I was initially confused by the 49.1% value. I would move the statistic down and incorporate somewhere after you have presented the context of having 638 sewersheds in total.

Line 176. I suggest that you first discuss that of the initial 638 presumed WWTP/sewershed, some were duplicates or decommissioned leaving 638-46=592 actual WWTP. This better highlights that you were able to create sewersheds for all active WWTPS/sewersheds.

Line 188. There are vast differences in population across NY so this isn’t surprising. Variation in the number of sewershed per county by population or population desnity would be interesting and illustrate why and where there are larger vs smaller systems. There needs to be a table summarizing the size and population distribution of sewersheds, perhaps stratified by urban/rural status of the county.

Line 194. Consider using ‘sewershed area’ rather than surface area.

Line 192 – 198. If the reported p-values are those reported by most software packages, the null hypothesis is that the correlation=0, s a p-value<0.05 indicates that the correlation is significantly different from 0, not a very meaningful statistical test. As these are descriptive statistics I recommend removing references to these p-values.

Figure 3 indicates that the correlations were between log(capacity) and sewershed characteristics (log base 10 I am assuming). However this paragraph and Table 2 imply that the values were not log transformed. Please clarify in the text and table. Please provide a rationale for the use of log(capacity). Non-household sewer inputs notwithstanding, there should be a direct correlation between the flow and population. As capacity is being used as a surrogate for flow, I think actual capacity should be used. If thee a rationale that I have missed, lease clarify.

Figure 2. Please label the color bar. Is yellow high density or low? I recommend reversing this so higher density population is represented by the darker color.

Figure 3. Revise the title to indicate that these are scatter plots, not correlations.

Discussion

Please address industrial and other non-household inputs and how they might be incorporated into this type of data system. In addition travel distance/travel time may be very helpful for understanding observed concentrations at the WWTP for non-conservative substances or infectious organisms that may decay in the system. This may be useful to note.

Line 213. The correlations did not involve the polygons, just permitted discharge capacity and population measures. Note my comments above regarding the use of log(capacity). Note that not all of the correlation coefficient would be considered high (0.57). It would be better to say that the majority of correlations were high, rather than saying ‘all’ and then saying in the next sentence that there was an exception.

I don’t agree that a high correlation between log of permitted discharge capacity provides evidence that the boundaries were accurate. It does perhaps show that the different methods may be comparable, however, there was likely relationships between the size of the system and the type of data used to create the sewersheds. This would be a useful result to present. At a minimum the text should consistently reflect that the correlations and relationships were related to the log of permitted capacity.

Line 273: It would be useful to see the distribution of populations in a table, as noted above. Did you get the same population estimates using the block group data from the ACS? These populations are unrealistically small and this represents about 10% of your sewersheds.

Line 279. Consider “These data will require updates in the future given the potential for …”…

There are a few items that readers would find useful. What was the effort need to complete this project? About how many hours were needed for large and small systems or by type of data provided? What is your assessment of accuracy and in what situations do you think that the boundaries are not reliable? How might the process be improved?

Further, the limitations of having only permitted discharge capacity (which may be different than actual discharge capacity/design capacity) needs to be addressed. It is greater than actual flow and this may vary with age of system and other factors. This affects the correlation coefficients. This is an important limitation to be discussed.

Reviewers' comments:

Reviewer's Responses to Questions

**Comments to the Author**

1. Does this manuscript meet PLOS Global Public Health’s publication criteria? Is the manuscript technically sound, and do the data support the conclusions? The manuscript must describe methodologically and ethically rigorous research with conclusions that are appropriately drawn based on the data presented.

Reviewer #1: Yes

Reviewer #2: Yes

2. Has the statistical analysis been performed appropriately and rigorously?

Reviewer #1: Yes

Reviewer #2: No

3. Have the authors made all data underlying the findings in their manuscript fully available (please refer to the Data Availability Statement at the start of the manuscript PDF file)?

Reviewer #1: Yes

Reviewer #2: Yes

4. Is the manuscript presented in an intelligible fashion and written in standard English?

Reviewer #1: Yes

Reviewer #2: Yes

5. Review Comments to the Author

Reviewer #1: This is a sampling protocol paper on how to use GIS files of sewer systems across jurisdictions for use in defining sewersheds for public health sampling, with a case study of New York presented. The paper is well written, practical, and innovative in documenting this approach.

1.) Abstract. Some areas of the United States do have centralized databases for geographic information on sewer systems. Do any other states have a fully centralized database? Add a qualifier to the statement “there are no centralized databases for geographic information on sewer systems”

2.) Line 40, add at the end of this line “or with the addition of pump stations.”

3.) Line 51, again need to qualify that you are talking just about NY here.

4.) Was there any overlap in service areas identified? This could be in high precipitation events for example where one sewershed diverts to another one.

5.) Were any of your sewersheds CSO/SSO?

6.) Authors should calculate high and low ranges for the counties studied of how many residents are on sewer, which indicates the total portion of the community captured by WBE samples.

7.) I think one of the most useful parts of this work is not the population covered, but knowing the spatial “gap” in households without services in areas where it “should” be available. For example, this might be an old house with a failing septic tank or straight pipe, which in addition to excluding this house from WBE samples also poses a risk to human health and the environment of surface water and your system could highlight these houses for public health attention. Can you choose one or two areas and conduct this additional analysis for comparison?

8.) In many older mid-western cities, straight pipes remain in pockets of urban areas. How do you account for this in NY, if present.

9.) A table in the supplement giving an overview of population size and a list of the counties you were/were not able to include would be helpful.

10.) Conclusion. How could the findings of this work be implemented across 2 or 3 jurisdictions? Is there a list of minimum information needed for this merge (Add a table)? What can be done within rural environments with limited digitization of sewer pipes (physical maps are only covered in the limitations)?

11.) Finally, when you submit the corrected version, please do check thoroughly, in order to avoid grammar, syntax or structure/presentation flaws.

Reviewer #2: This is a very useful paper detailing a project in NY state delineating the boundaries of sewersheds using a variety of extant data. This paper would be very helpful for practitioners and researchers and the resulting sewershed data has been valuable for public health actions and research. The manuscript is clear and the methods area well described; Fig 1 is very nice.

I have two primary comments. First, I do not agree that the correlations between log(capacity) and population metrics provide a high degree of evaluating accuracy. I have provided more detailed comments below. If the correlations are with capacity and not log(capacity), I apologize for my misunderstanding.

I imagine that the authors have a great deal of knowledge about how accurate the resulting sewersheds are and the factors that result in having accurate or inaccurate boundaries. This leads to my second comment: this experience and a few other topics would be of great use to readers and should be covered in he discussion.

I have provided more detailed comments addressing these two points below.

Specific comments.

Background

Line 37 – Consider replacing ‘area’ to ‘extent’

Line 42 – Note that lift stations are also used to not only feed force mains, but to feed gravity flow sewers from a higher starting elevation.

Methods

Line 84 – where did the special district tables come from?

Line 143: Did you use the same dasymetric apportionment for he block groups as you used for the 2010 block populations? Please clarify.

Line 154: It doesn’t seem that this process was validation as you didn’t appear to have a ‘true’ or gold standard boundary to compare to other derived boundaries, and when you did have a ‘good’ boundary, you used that. In some cases it sounds like you used a combination of methods. In addition it appears that you compared methods to get a sense of the reliability of your estimates. Comparing population estimates, which have inherent and non-random biases due to the apportionment methods, to the permitted flow may be a form of validation. However permitted flow is not actual average flow (which we have been able to get from most WWTPs). Were there any criteria used to determine if a sewershed appeared to be accurate based on a relationship between population and permitted flow?

This is a relatively small point. I recommend that you re-word the section and characterize it as just QA. Reliability is a concept you seem to be describing in this section. Quality Control is more accurate as you are assessing the resulting product (the sewershed boundaries) rather than the efforts and protocols set up in the process of creating the sewersheds to assure high quality boundaries.

Results

Line 170. I was initially confused by the 49.1% value. I would move the statistic down and incorporate somewhere after you have presented the context of having 638 sewersheds in total.

Line 176. I suggest that you first discuss that of the initial 638 presumed WWTP/sewershed, some were duplicates or decommissioned leaving 638-46=592 actual WWTP. This better highlights that you were able to create sewersheds for all active WWTPS/sewersheds.

Line 188. There are vast differences in population across NY so this isn’t surprising. Variation in the number of sewershed per county by population or population desnity would be interesting and illustrate why and where there are larger vs smaller systems. There needs to be a table summarizing the size and population distribution of sewersheds, perhaps stratified by urban/rural status of the county.

Line 194. Consider using ‘sewershed area’ rather than surface area.

Line 192 – 198. If the reported p-values are those reported by most software packages, the null hypothesis is that the correlation=0, s a p-value<0.05 indicates that the correlation is significantly different from 0, not a very meaningful statistical test. As these are descriptive statistics I recommend removing references to these p-values.

Figure 3 indicates that the correlations were between log(capacity) and sewershed characteristics (log base 10 I am assuming). However this paragraph and Table 2 imply that the values were not log transformed. Please clarify in the text and table. Please provide a rationale for the use of log(capacity). Non-household sewer inputs notwithstanding, there should be a direct correlation between the flow and population. As capacity is being used as a surrogate for flow, I think actual capacity should be used. If thee a rationale that I have missed, lease clarify.

Figure 2. Please label the color bar. Is yellow high density or low? I recommend reversing this so higher density population is represented by the darker color.

Figure 3. Revise the title to indicate that these are scatter plots, not correlations.

Discussion

Please address industrial and other non-household inputs and how they might be incorporated into this type of data system. In addition travel distance/travel time may be very helpful for understanding observed concentrations at the WWTP for non-conservative substances or infectious organisms that may decay in the system. This may be useful to note.

Line 213. The correlations did not involve the polygons, just permitted discharge capacity and population measures. Note my comments above regarding the use of log(capacity). Note that not all of the correlation coefficient would be considered high (0.57). It would be better to say that the majority of correlations were high, rather than saying ‘all’ and then saying in the next sentence that there was an exception.

I don’t agree that a high correlation between log of permitted discharge capacity provides evidence that the boundaries were accurate. It does perhaps show that the different methods may be comparable, however, there was likely relationships between the size of the system and the type of data used to create the sewersheds. This would be a useful result to present. At a minimum the text should consistently reflect that the correlations and relationships were related to the log of permitted capacity.

Line 273: It would be useful to see the distribution of populations in a table, as noted above. Did you get the same population estimates using the block group data from the ACS? These populations are unrealistically small and this represents about 10% of your sewersheds.

Line 279. Consider “These data will require updates in the future given the potential for …”…

There are a few items that readers would find useful. What was the effort need to complete this project? About how many hours were needed for large and small systems or by type of data provided? What is your assessment of accuracy and in what situations do you think that the boundaries are not reliable? How might the process be improved?

Further, the limitations of having only permitted discharge capacity (which may be different than actual discharge capacity/design capacity) needs to be addressed. It is greater than actual flow and this may vary with age of system and other factors. This affects the correlation coefficients. This is an important limitation to be discussed.

6. PLOS authors have the option to publish the peer review history of their article (what does this mean?). If published, this will include your full peer review and any attached files.

**Do you want your identity to be public for this peer review?** For information about this choice, including consent withdrawal, please see our Privacy Policy.

Reviewer #1: No

Reviewer #2: No

---

## [Editor Report · Decision Letter 1]

12 Dec 2022

Using geographic information systems to link population estimates to wastewater surveillance data in New York State, USA

PGPH-D-22-01363R1

Dear Dr. Hill,

We are pleased to inform you that your manuscript 'Using geographic information systems to link population estimates to wastewater surveillance data in New York State, USA' has been provisionally accepted for publication in PLOS Global Public Health.

Best regards,

Thomas P. Van Boeckel

Academic Editor